# Assessment of Deoxynivalenol in Wheat, Corn and Its Products and Estimation of Dietary Intake

**DOI:** 10.3390/ijerph17155602

**Published:** 2020-08-03

**Authors:** Shahzad Zafar Iqbal, Sunusi Usman, Ahmad Faizal Abdull Razis, Nada Basheir Ali, Tahmina Saif, Muhammad Rafique Asi

**Affiliations:** 1Department of Applied Chemistry, Government College University Faisalabad, Faisalabad 38000, Pakistan; tahminasaif27@gmail.com; 2Institute of Bioscience, Universiti Putra Malaysia, 43400 UPM Serdang, Selangor, Malaysia; usunusi.bch@buk.edu.ng; 3Department of Food Science, Faculty of Food Science and Technology, Universiti Putra Malaysia, 43400 UPM Serdang, Selangor, Malaysia; nada44basher@gmail.com; 4Plant Production Division, NIAB, Faisalabad 38000, Pakistan; mrasi@niab.edu.pk

**Keywords:** DON, wheat, corn, dietary estimation, wheat flour, corn flour

## Abstract

The main goal of the present research was to explore the seasonal variation of deoxynivalenol (DON) in wheat, corn, and their products, collected during 2018–2019. Samples of 449 of wheat and products and 270 samples of corn and their products were examined using reverse-phase liquid chromatography with a UV detector. The findings of the present work showed that 104 (44.8%) samples of wheat and products from the summer season, and 91 (41.9%) samples from winter season were contaminated with DON (concentration limit of detections (LOD) to 2145 µg/kg and LOD to 2050 µg/kg), from summer and winter seasons, respectively. In corn and products, 87 (61.2%) samples from summer and 57 (44.5%) samples from winter season were polluted with DON with levels ranging from LOD to 2967 µg/kg and LOD to 2490 µg/kg, from the summer and winter season, respectively. The highest dietary intake of DON was determined in wheat flour 8.84 µg/kg body weight/day from the summer season, and 7.21 µg/kg body weight/day from the winter season. The findings of the work argued the need to implement stringent guidelines and create awareness among farmers, stakeholders, and traders of the harmful effect of DON. It is mostly observed that cereal crops are transported and stockpiled in jute bags, which may absorb moisture from the environment and produce favorable conditions for fungal growth. Therefore, these crops must store in polyethylene bags during transportation and storage, and moisture should be controlled. It is highly desirable to use those varieties that are more resistant to fungi attack. Humidity and moisture levels need to be controlled during storage and transportation.

## 1. Introduction

Cereal plays a vital role in human health and wellbeing. Worldwide, cereals are the most important source of energy, and in developed countries, about 30% of daily calories are derived from cereals, compared to 60 to 80% calories in developing countries [1]. Pakistan is placed seventh in wheat production (25.07 million tons), 10th in rice production (10.8 million tons) and produces 6.31 million tons of corn [2]. In Pakistan, wheat, rice, and corn are the main staple food crops. Pakistan has extreme weather conditions, due to its presence in the subtropical region. Furthermore, unseasonal rains, monsoons rains, and flash floods are the main factors that provide favorable conditions for the proliferation of fungi in food and food products [3]. The use of traditional cultivation practices and unawareness of good harvesting and storage practices are other main factors that provide fungal attacks in crops along the food chain [4].

Mycotoxins are an important class of food toxins, and it is classified as naturally occurring secondary metabolites [5]. The fungi such as *Fusarium*, *Aspergillus*, and *Penicillium* are recognized as major producer of mycotoxins [6,7]. The diversity and variation in mycotoxins could be imagined, because 450 different types of structure of such toxins are identified and classified [8]. The most studied mycotoxins are aflatoxins, ochratoxin A, deoxynivalenol, zearalenone, and fumonisin, due to their toxic nature and adverse effects on food quality and safety [9]. The deoxynivalenol (DON) mycotoxin is also known as vomitoxin is closely related to trichothecenes [10]. The fungi, such as *Fusarium culmorum* and *Fusarium graminearum* are mainly responsible for producing this type of mycotoxin (deoxynivalenol) [11]. The studies have shown that at the molecular level, it binds to the ribosome and inhibiting protein synthesis and thus disrupts normal cell function [12]. The low doses of DON exert toxic effects, including immune dysfunction and growth impairment. However, lethal dose exposure can lead to leukocytosis, hemorrhage, diarrhea, endotoxemia, and shock-like death [13].

In order to avoid the health hazard in the local population, strict regulations are imposed for DON in wheat and wheat products used for the consumption of humans and animals. The permissible limit of 2 mg/kg of DON in raw wheat, barley, and maize has been implemented by the Codex Alimentarius Commission (CAC). In contrast, the level of 2000 μg/kg has been established by the European Union for different types of unprocessed wheat and oat for different populations. Similarly, there is a permissible legal limit of 1000 μg/kg for raw or those cereals which would be unprocessed, however, a limit of 750 μg/kg for those foodstuffs intended for consumers and dry pasta and 200 μg/kg for snacks and breakfast cereal proposed for the consumption of baby food [14].

In Pakistan, wheat is mostly used to prepare bread, and a variety of traditional and culturally flatbread are famous and prepared in a hot clay oven known as “tandoor,” which is common throughout rural and urban areas of Pakistan. The country’s wheat consumption per capita was 124 kg per person per year, and 72% of calorie intake comes from wheat flour, one of the highest in the world [15]. The high incidence of some other mycotoxins in cereal crops from our previous reports, i.e., 30% samples of wheat products were found to be contaminated with aflatoxins, ranging from limit of detections (LOD) to 69.6 µg/kg, and 31% samples of wheat products were found to be contaminated with zearalenone ranging from LOD to 69.8 µg/kg [16]. In another study 52% cereal products were found to be contaminated with aflatoxins with concentration ranging from LOD to 9.95 µg/kg, 50% cereal products were found to be contaminated with ochratoxin A (LOD to 9.60 µg/kg), and 56% samples of cereal products were found to be contaminated with zearalenone (LOD to 110.45 µg/kg) [17]. It is worth mentioning that we have studied the incidences of AFs, OTA, ZEN in cereals and, recently, fumonisin B_1_ in wheat and barley [18]. Therefore, the current research was planned to investigate the presence of DON in most staple food crops of Pakistan, i.e., wheat and corn. The undertaken research was focused on:(1)Examining the amount of DON in wheat, corn and their products from summer and winter seasons;(2)Relating the concentrations of DON with EU recommended limits;(3)Estimating the dietary intake of DON in wheat and corn products.

A few reports are present on the occurrence of DON in wheat or corn products from Punjab, Pakistan [19,20] and from Turkey [21]. The findings of the present work will help to generate data and, thus, help law enforcement agencies to establish strict regulations for this toxin.

## 2. Materials and Methods

### 2.1. Samples

The 449 samples of wheat and products (wheat flour, semolina, porridge, wheat bread) (232 from the summer season, and 217 from winter season) and 270 samples of corn and products (corn flour, cornflakes, boiled corn, corn bread) (142 samples from the summer season and 128 samples from winter season) were collected from various shops, supermarkets and stores of the central cities of Punjab (Lahore, Faisalabad, Jhang and Shorkoat), Pakistan, during December 2017 to June 2018. The samples of wheat, corn and their products were purchased randomly. The summer season is composed from May to August, and the winter season is from November to January. The 1 kg of sample size of each crop and processed food was maintained. The samples not in ground form were grounded using grinding mill (Retsch, ZM 200, Haan, Düsseldorf, Germany). The samples were taken in polyethylene plastic bags and stored in the freezer at −20 °C, until further analysis.

### 2.2. Chemicals and Reagents

The regents like polyethylene glycol 8000 (PEG), high purity acetonitrile, and methanol (≥98% purity), were obtained from Sigma-Aldrich (Sigma-Aldrich, Lyon, France), however, the standard of DON (100 mg/mL in ACN), were purchased from Sigma Aldrich (Saint-Louis, MO, USA). The linearity curve was constructed by making different levels of DON, i.e., 200, 400, 600, 800, 1600, 2000, and 2600 µg/L in acetonitrile, and each vial was labeled and stored in an appropriate place in refrigerator at −20 °C. In the undertaken study, other chemicals and reagents used were of high purity.

### 2.3. Extraction of DON and HPLC Conditions

DON was extracted from cereal products following the method as documented by [22]. Briefly, in 20 mL ultra-pure water, a 5 g sample was homogenized in 50 mL Teflon tubes and centrifuged at 6500 rpm for 1 min by adding 1 g of PEG. Then, 200 mL water was applied on dry bread to regain its moisture content, before it was homogenized for 5 mint at room temperature at 8000 rpm. The extract was filtered with filter paper after centrifugation, and 5 mL of the filtrate was passed to the immunoaffinity column (IAC) (VICAM, Watertown, MA, USA) for DON. After that, the column was washed twice with 10 mL purified water each time, and 1 mL pure methanol was used to extract DON from the IAC column. Then, the elute was diluted with purified water two times, i.e., 0.5 mL elute: 0.5 mL water. Finally, the diluted extract was subjected to HPLC analysis, before passing it through 0.22 µm nylon syringe filters. The HPLC system (Shimadzu, Kyoto, Japan) had a separation C18 Supelco column (4.6 × 250 × 5 mm) (Discovery HS, Bellefonte, PA, USA) and a UV detector (RF-530). The mobile phase has consisted of a mixture of 30% methanol and 70% of water. The 1.2 mL/min of flow rate was set off the mobile phase. The 50 μL samples were injected into the column for analysis, the column temperature was maintained at 30 °C, and the detection wavelength was 218 nm.

### 2.4. Dietary Intake Evaluation

The analysis of the dietary intake of DON in cereal products, a previous method Alim et al. [17], was followed. The consumption data was gathered by providing a food frequency questionnaire around 500 individuals and asked them about the cereal products they used in the last 2 weeks. The questionnaire is completed in every respect by memorizing the cereal products used, its quantity, frequency, and other food ingredients. The weight of normal cup, bowl, and average size plate was also mentioned, and based on this information, the exact weight of food consumed was estimated. The average body weight of and individual was proposed at 60 kg. The dietary intake was estimated as
(1)$$\mathrm{Dietary}\;\mathrm{Intake}\;\mathrm{of}\;\mathrm{Don}\;(\mathrm{ng}\;{\mathrm{kg}}^{-1}\;{\mathrm{day}}^{-1})=\frac{Consumption\;of\;cereals\;\left(g\right)x\;Mean\;level\;of\;DON\;\left(µ\frac{g}{kg}\right)}{Average\;weight\;of\;individual\;\left(kg\right)}$$

### 2.5. Method Validation

The HPLC quality control parameters like linearity, precision, limit of detections (LOD), limit of quantifications (LOQ), repeatability, and reproducibility of DON were calculated. The LOD was determined as 3:1 of signal to-noise ratio and LOQ was calculated as 10:1 signal-to-noise ratio [23]. The repeatability and reproducibility of the method were determined by spiking 2 different amounts of DON (5 replicate each) in a sample on the same day. Repeatability and reproducibility described in current research R^2^ ≥ 0.998, which shown good linear function. The recoveries were determined by adding a known level of DON, i.e., 100, 200, 400, 800, 1600, and 2000 µg kg^−1^ were added in matrix samples. The recoveries ranged between 80.8 to 92.2% with relative standard deviation from 9 to 27%, as represented in Table 1.

### 2.6. Statistical Analysis

The analysis of DON were statistically observed and represented as average levels and standard deviations. However, the samples below the detection limits, but greater than zero were substituted with $\frac{LD}{\sqrt{2}}$, and the (R^2^) was analyzed using linear regression/correlation analysis. The significant difference in the levels of DON among different seasons was calculated using a one-way analysis of variance (SPSS, IBM, Statistics 21).

## 3. Results and Discussion

### 3.1. Method Validation

The recoveries of DON in wheat, corn, wheat and corn flour are represented in Table 1, and the recoveries were greater than 80%, as recommended by (AOAC method 991.31). The recovery analysis were done in different mycotoxins with concentrations of 6.5 and 13 ng/g for AFs, 10 and 20 ng/g for OTA, and 100 and 200 ng/g for ZEA were between 74.1% and 104.8%, and the relative standard deviation (RSD) values were in the range of 2.4 to 11.9% [24]. Furthermore, the recoveries were within the range of 70–110%, as required by European Commission regulation 401/2006 [25], hence, they were considered satisfactory. However, the recoveries of corn were comparatively greater, as compared to the wheat samples. This might be due to the high content of polar components in corn matrix, which dissolve polar solvent more easily, and further investigation in this regard is highly needed. Form China, the detection of LOD and LOQ of DON in wheat grain were 0.5 µg/kg, and 1.5 µg/kg, much lower compared to current study [23]. The lower LOD and LOQ were determined in wheat of DON, i.e., 11.3 and 37.6 μg/kg, respectively [26]. Thus, LOD and LOQ are two quantities that have accuracies that depend on the sensitivity of the device used [27].

### 3.2. DON Incidence in Wheat and What Products

The incidence and occurrence levels of DON were investigated in 232 samples of wheat and products from the summer season and in 217 samples from the winter season, as represented in Table 2. The highest average amount of DON was present in wheat flour, i.e., 1325.5 ± 24.7 µg/kg samples from the summer season and the lowest mean level was found in wheat porridge (934.1 ± 19.2 µg/kg) from the summer season. However, the highest mean levels of DON from the winter season were 1080.7 ± 30.5 µg/kg in wheat flour samples. The results have shown that 104 (44.8%) samples out of 232 from the summer season were found to be contaminated with DON in wheat and wheat products, and 91 (41.9%) out of 217 samples were found to be positive with DON from the winter season. The results of DON in summer and winter samples of wheat and wheat products were statistically significant at α 0.05, except in wheat (type 1) samples, which were nonsignificant at α 0.05. The samples were 54 (23%) from summer, and 20 (9.2%) samples from winter of wheat, and wheat products had levels of DON higher than EU’s recommended limits, as shown in Figure 1.

Previous studies have shown very high incidence as well as occurrence levels of DON in wheat samples. Bryla et al. [28] from Poland has documented that 100% of wheat samples were found to be positive, with DON with a mean level of 770.7 µg/kg, ranging from 82 to 2975 µg/kg. In another study from China, 87.5% samples of wheat and wheat products were found to be positive, with DON with a range of 12.5 to 1920.4 µg/kg, 100% samples of wheat flour, the level of DON was 51.6 to 1308.9 μg/kg, and in Chinese steamed bread, the mean level was 54.5–845.4 μg/kg [11]. Palacios et al. [7], from Argentina, has shown that DON was found in all samples of wheat with concentrations varying between LOD (50 µg/kg) to 9480 µg/kg. Liu et al. [29], from China, have revealed that out of 672 samples of wheat, 91.5% of samples were found to be positive with DON, at levels ranging from 2.4 to 1130 µg/kg. In another study, Ji et al. [30] have detected the levels of DON in 180 wheat samples from China, with levels ranging from 14.52 to 41157.13 µg/kg (mean level of 488.02 µg/kg). The hot and humid environmental conditions might explain the high incidence levels of DON in wheat and wheat products samples from the summer season during summer in Pakistan. The levels of toxins during seasons depend on the variation in the structure of toxins during cropping seasons [31]. The crop of wheat is harvested in summer season mostly in May and June, which might increase the probabilities of fungal attack during pre-harvest, or postharvest, or during the storage of mycotoxins production. The other factors of a high incidence of DON in wheat and wheat products might be the use of those wheat varieties which were sensitive for fungal attack, old traditional farming practices, no crop rotation, and no-till farming. Furthermore, in villages, people store wheat or corn in mud bins, which may absorb moisture from the environment and cause fungal attacks [16]. Furthermore, temperature, humidity, sunlight effect, and survival of mycotoxigenic fungi to host are essential factors [32].

### 3.3. Corn and Products

The incidence and occurrence levels of DON were investigated in 142 samples of corn and products from the summer and 128 samples from winter seasons, as revealed in Table 3. The results have documented that 87 (61.2%) samples out of 142 were found positive from the summer season, and 57 (44.5%) samples from the winter season were found contaminated with DON. The highest mean level of 1434.8 ± 25.5 µg/kg in corn flour samples from summer season was found, and the lowest mean level of DON was found in cornbread 620.8 ± 17.8 µg/kg, from the winter season. The incidence levels of DON in corn and corn products from the summer season were statistically significant from the samples of winter season (α 0.05) except cornbread samples. The samples of corn and corn products having levels of DON higher than the recommended limits of the EU are compared in Figure 2.

The levels of DON contamination (ranged, LOD to 2967 µg/kg in summer and LOD to 2490 µg/kg in winter season) found in the corn and corn products are much higher in undertaken research. In other studies, the levels of 57 to 825 µg/kg in corn flour samples from Tanzania [33], and 43 to 435 µg/kg stated for corn in Cameroon [34] and in Italy, where levels of 1 to 930 µg/kg are described in processed food products in the retail market [35]. The results are comparable to the levels (up to 1250 µg/kg) that were documented in those processed food used in other countries [30]. Higher incidences have been documented by Ji et al. [11] from China, where 73.3% of corn flour samples were contaminated with DON, with a range of 26.4 to 138.2 μg/kg. Kamala et al. [33] from Tanzania, collected corn kernels from 300 household stores, and found that 63% of samples were contaminated with DON, ranging from 68 to 2196 μg/kg, in agreement with the findings of the undertaken research. Similarly, a very high incidence, i.e., 85% of corn samples, was found to be contaminated with DON in Croatia, with a maximum of 17.92 mg/kg (µg/kg). The factors like high humidity, high moisture content and high temperature might explain the high levels of DON in corn (Pleadin et al. [36]). Setyabudi et al. [37], from Indonesia, documented that 26 corn based-food products have levels of DON, ranging from 103.4 to 170.3 μg/kg in corn flour, 62.6 to 308.3 μg/kg in extruded corn, 59.9 to 202.1 μg/kg in popcorn, and 67.1 to 348.0 μg/kg fried corn, respectively. Martos et al. [38], from Canada, investigated 15 samples of corn for the occurrence of DON, and 14 samples were found to be positive, with a mean level of 1513.5 µg/kg (ranged from 574 to 4865 µg/kg). The variation and diversity in the levels of mycotoxins in cereal crops might be due to the origin and the year of harvesting crops [39]. However, fungi proliferation and ultimately, the production of mycotoxins in cereal crops might be the result of climatic and improper storage conditions [3].

### 3.4. Dietary Estimation

The dietary intake of DON in samples of wheat and corn products from summer and winter seasons were investigated and presented in Table 4. The highest dietary intake of DON was found in wheat flour, i.e., 8.8 µg/day kg bodyweight from the summer season. The lowest dietary intake was found in semolina, i.e., 1.37 µg/day kg body weight in the summer season. However, the dietary intake level of 7.21 was found in wheat flour samples from the winter season. Raad et al. [40] estimated the dietary intake of DON in breads and toasts was 1052 ng kg-1 body weight day-1 (1.05 µg/day kg bodyweight), which is much lower than the dietary intake of the present study. Similarly, a much lower dietary intake was documented by Cano-Sancho et al. [41] in pasta samples, which was 27 to 38 ng kg^−1^ body weight day^−1^. In another study, the 95th percentile exposure of total DON forms was determined in children, from 648 to 1030 ng/kg bw/day (LB/lower bound/and UB/upper bound), in women from 272 to 490 ng/kg bw/day, and in men from 362 to 923 ng/kg bw/day [42]. In Romania, the dietary exposure of DON in wheat-based products for direct human consumption was 669 ng/kg bw/day (LB), and 690 ng/kg bw/day (UB) in the adult population [43]. However, in Portugal the mean of daily intake of DON was 53.9 ng/kg bw/day in children under three years of age, through the consumption of three groups of cereal based products (breakfast cereals, infant cereals and biscuits) [44].

## 4. Conclusions

The incidence and contamination of DON wheat and corn products were observed to be relatively higher, as likened with the permissible allowed limits of the EU. To our best knowledge, very few reports were documented on the estimation of DON in wheat and corn products from Pakistan. The current results will be helpful for food authorities to establish strict regulations for this toxin. The results will also help to create awareness among traders, farmers, and local consumers. It is mostly observed that cereal crops are transported and stockpiled in jute bags, which may absorb moisture from the environment and produce favorable conditions for fungal growth. Therefore, these crops must store in polyethylene bags during transportation and storage, and moisture should be controlled. It is highly desirable to use those varieties which are more resistant to fungi attack. Humidity and moisture levels need to be controlled during storage and transportation.

## Figures and Tables

**Figure 1 ijerph-17-05602-f001:**
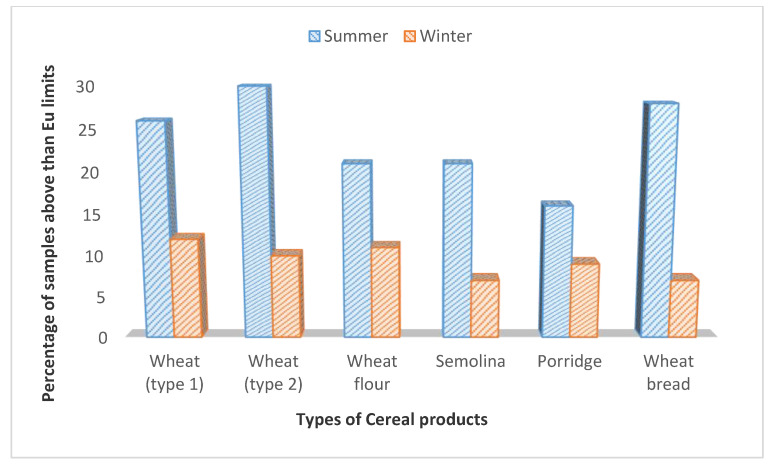
Comparision of sample of wheat and products having levels of DON higher than EU limits.

**Figure 2 ijerph-17-05602-f002:**
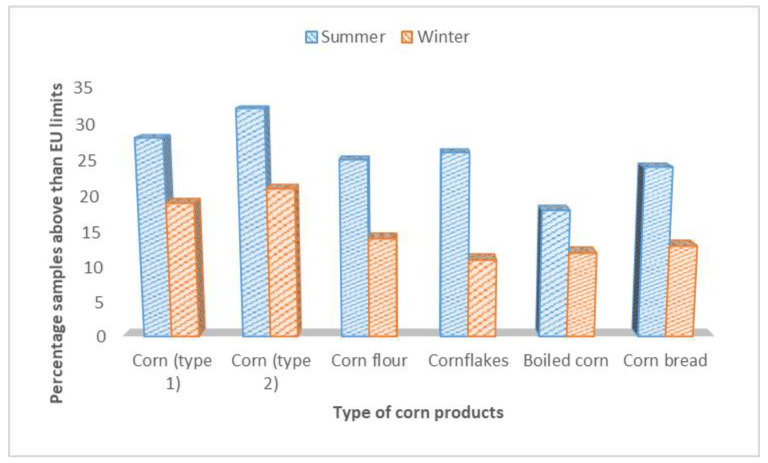
Comparison of sample of corn and products having levels of DON higher than EU limits.

**Table 1 ijerph-17-05602-t001:** Recovery percentage of deoxynivalenol (DON) in wheat, corn, wheat flour and corn flour samples.

Don Level	Wheat		Wheat Flour		Corn		Corn Flour	
µg/Kg	Recovery (%)	RSD (%)	Recovery (%)	RSD (%)	Recovery (%)	RSD (%)	Recovery (%)	RSD (%)
100	82.5	14	80.8	9	87.4	10	88.9	13
200	87.8	18	85.4	12	89.8	16	90.0	15
400	89.0	19	84.7	19	92.6	20	86.7	10
800	90.0	10	88.9	10	88.9	25	85.6	18
1600	87.8	23	84.1	19	86.7	25	86.7	20
2000	89.0	26	82.0	21	89.5	27	83.5	27

RSD = relative standard deviation; LOD = 50 µg/kg; LOQ = 100 µg/kg.

**Table 2 ijerph-17-05602-t002:** Incidence and occurrence of DON in wheat and products collected during summer and winter seasons from different cities of Punjab, Pakistan.

Samples Type	Summer				Winter			
	Samples	Positive	Mean	Range	Samples	Positive	Mean	Range
	*n*	*n* (%)	µg/kg ± S.D	µg/kg	*n*	*n* (%)	µg/kg ± S.D	µg/kg
Wheat (type 1)	34	13 (38.2)	1083.5 ± 15.2 ^a^	LOD-2145	25	11 (44.0)	1020.8 ± 21.6 ^a^	LOD-1945
Wheat (type 2)	40	21 (52.5)	945.1 ± 18.9 ^b^	LOD-2021	40	15 (37.5)	790 ± 20.9 ^c^	LOD-2050
Wheat flour	38	21 (55.2)	1325.5 ± 24.7 ^c^	LOD-1890	38	15 (39.5)	1080.7 ± 30.5 ^a^	LOD-1500
Semolina	42	18 (42.8)	1026.4 ± 17.3 ^a^	LOD-1459	41	18 (43.9)	750. 7 ± 25.6 ^b^	LOD-1130
Wheat porridge	49	20 (40.8)	934.1 ±19.2 ^b^	LOD-980	46	19 (41.3)	690. 5 ± 19.6 ^c^	LOD-950
Wheat bread	29	11 (37.9)	990.4± 21.8 ^b^	LOD-1120	27	13 (48.1)	705.6 ± 23.7 ^c^	LOD-1080
Total	232	104 (44.8)		LOD-2145	217	91 (41.9)		L0D-2050

The data in parenthesis represents the percentage to total analyzed samples. LOD = limit of detection (LOD 50 µg/kg) The mean levels with different English alphabetic within the row indicated statistically significant results (*p* ≤ 0.05). Type 1 and Type 2 represents different varieties.

**Table 3 ijerph-17-05602-t003:** Incidence and occurrence of DON in corn and products from summer and winter seasons from different cities of Punjab, Pakistan.

	Winter				Summer			
Samples Type	Samples	Positive	Mean ± S.D	Range	Samples	Positive	Mean ± S.D	Range
	*n*	*n* (%)	µg/kg	µg/kg	*n*	*n* (%)	µg/kg	µg/kg
Corn (type 1)	25	14 (56.0)	1345.7 ± 22.3 ^a^	LOD-2967	20	9 (45.5)	978. 7 ± 15.6 ^a^	LOD-2490
Corn (type 2)	26	16 (61.5)	998.2 ± 11.2 ^b^	LOD-2234	25	11(44.0)	680.6 ± 11.8 ^c^	LOD-2650
Corn flour	25	18 (72.0)	1434.8 ± 25.5 ^a^	LOD-2434	26	12 (46.1)	860.7 ± 13.9 ^b^	LOD-1460
Cornflakes	24	12 (50.0)	1245.9 ± 18.5 ^a^	LOD-1534	22	10 (45.4)	910.4 ± 23.7 ^b^	LOD-1090
Boiled corn	22	13 (59.5)	1067.5 ±14.1 ^b^	LOD-920	20	9 (45.0)	850.9 ± 15.7 ^d^	LOD-980
Corn bread	20	14 (70.0)	640.5± 11.3 ^c^	LOD-1320	15	6 (40.0)	620.8 ± 17.8 0^c^	LOD-1250
Total	142	87 (61.2)		LOD-2967	128	57 (44.5)		LOD-2490

The data in parenthesis represents the percentage to total analyzed samples. LOD = limit of detection (LOD 50 µg/kg). The mean levels with different English alphabetic within the row indicated statistically significant results (*p* ≤ 0.05). Type 1 and Type 2 represents different varieties.

**Table 4 ijerph-17-05602-t004:** Dietary exposure assessment for DON in wheat, corn and products consumed by local population.

		Summer		Winter	
Cereal Product	Per Capita Consumption	DON Mean Level	Dietary Intake	DON Mean Level	Dietary Intake
	g day^−1^	ng g^−1^	µg day^−1^ kg^−1^ Bodyweight	(ng g^−1^)	µg day^−1^ kg^−1^ Bodyweight
Wheat flour	400	1325.5	8.84	1080.7	7.21
Bread	400	990.4	6.61	705.6	4.71
Semolina	80	1026.4	1.37	750.7	1.01
Porridge	90	934.1	1.41	690.5	1.04
Corn flour	200	1434.8	4.78	860.7	2.87
Bread	200	640.5	2.14	620.8	2.07
Cornflakes	80	1245.9	1.66	910.4	1.24
Boiled corn	200	1067.5	3.56	850.9	2.84

The average weight = 60 (kg).

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
