# Peer review of "Assessment of Deoxynivalenol in Wheat, Corn and Its Products and Estimation of Dietary Intake"

_ijerph, 2020, doi:10.3390/ijerph17155602_

Round 1

Reviewer 1 Report

Although a moderate English was indicated, the authors might want to take another look at the 'English" presentation in the manuscript.

Abstract: The authors should endeavor to indicate some control measures that could be suggested to farmers, stakeholders and traders on probable ways to mitigate mycotoxin contamination.

Line 28: Authors should please try to elucidate more on the importance of cereals here, highlight their role as staples and one of the primary sources of calories/energy intake. 

Lines 29 and 30: Refs 1 and 2 should be deleted and information should rather be sourced from FAOSTAT (http://www.fao.org/faostat/en/#home).

Line 35: ... in crops, along the food chain.

Line 37-38: Please provide reference.

Line 38: Fusarium, Aspergillus, and Penicillium are not the only fungi genera producing mycotoxins. Authors should either indicate them as being the major ones or provide all the conventional fungi genera.

Line 40: aflatoxins,

line 40-41: Authors should add funmonisins to this list.

Line 43-44: Which type of mycotoxin is being reffered to here?

Line 59: Please provide details of this high incidence. What is the maximum level recorded and what % incidence?

Line 66: Please provide references for these reports from Pakistan. This could also be extended to other neighbouring countries.

Section 2.1: What were the criteria for sampling the reported products? The authors should please describe the 'products' being referred to here. What products were these? How are they obtained? Were the products derived from the same raw materials used to obtain the products reffered to here?

Line 89: Were the samples milled/communited before being 'homogenized'.

Line 113-114: LOD and LOQ. How were LOD and LOQ calculated?

Line 118-119: Were the subsequent values reported here based on the recoveries recorded?

Line 128-129: This sentence is quite confusing and authors would need to recast it to convey the intended meaning. "However, the samples' blew detection limits, but grated than zero were substituted with, and the (R2) was analyzed linear regression/correlation analysis"?

Section 3: Authors might might to start this section with a paragraph on method validation. Are the recovery rates within the recommended ranges. What could be responsible for the slightly higher recovery rate in maize as compared to wheat?

Line 135: DON.

Line 136: Please see my comment on section 2.1 and please redress. Where was the porridge from. How was it processed/prepared? Was it from wheat or from maize? This is the first mention of 'porridge' and readers might be confused.

Line 138-139: found to be contaminated with DON....

Authors should endeavor to compare wheat and maize samples as well as the products derived from them. How well has processing assited in reducing DON levels?

Conclusion: Possible control measures? Fumonisins (FB1, FB2 and FB3) are known to majorly contaminate cereals, especially maize, but was not analyzed in this study. Could there be a particular reason for this?

Conclusion: It would be best to indicate in the conclusion that further studies should also look into a number of mycotoxins that frequently contaminate cereals, as well as emerging ones.

Reviewer 2 Report

The manuscript by Iqbal and co-workers describe an analysis of contamination of wheat, corn and their products from Pakistan market by deoxynivalenol. The authors analyzed over 700 samples and evaluated dietary intake of DON by local population. The article is the first considering DON assessment on population of Pakistan, however it will require some corrections before the publication.

Comments:

  • The authors could easily expand their research by the analysis of nivalenol and masked forms of deoxynivalenol (e.g. DON-3-G) using different immunoaffinity column (DON-NIV by Vicam). It would provide much more information on dietary intake of mycotoxins. 
  • Line 113-114: abbreviation of limit of detection is LOD, while limit of quantification is LOQ
  • minor typing errors through the text
  • Table 1. lack of parentheses
  • Table 2. what is the difference between type 1 and type 2 of wheat ? please explain
  • Table 2. standard deviations presented in the table look surprisingly low considering minimal and maximal values; could you please check your calculations
  • Table 3. what is the difference between type 1 and type 2 of corn ? please explain
  • Table 3. standard deviations presented in the table look surprisingly low considering minimal and maximal values; could you please check your calculations
  • Line 218. Please expand discussion on dietary exposure estimation using more data from the literature
  • Table 4. Please rearrange the table, so it would be more organized.

Reviewer 3 Report

The authors present their results on assessing the DON concentrations in food products of Pakistan. The methods, results

Maybe the authors should add the origin and year of the samples to the abstract.

The authors should read the publication once more carefully since some minor spelling mistakes are still present.

The authors should include recent publications by the working group of Dänicke, Friedrich-Löffler-Institute Braunschweig since they conducted some major research on DON, and their results would be a very good basis to discuss your results. Especially for risk assessment.

Finally, the manuscript is well written, and the method seems to be sound. If the authors would improve their discussion and the risk assessment part, the reviewer would suggest accepting the publication.

Round 2

Reviewer 1 Report

Abstract: Although the authors indicated to have provided control measure in the conclusion, this should still be highlighted in the abstract. Knowing that the abstract is usually what is first seen in a scientific article, it would be ideal to do so.

Line 57-65: The reports of studies including highlighting ZEA and OTA as part of the mycotoxins investigated raising questions as to why only DON was done in this study. The authors should please redress this, perhaps by providing an explanation in the latter part of the introduction as to why only DON was investigated.

Line 121-122: Please provide a reference for this.

Line 135: I still dont think a proper English edit of this manuscript was done. Can samples possibly 'blow' detection limits? "the samples' blew detection limits"???

Section 3.1: Please provide adequate references for studies rather than an "AOAC method". There are EU guidelines for this as well as recent studies in this regard. Furthermore, what does literature say regarding LOQ and LOD values and what was observed in this study?

Line 251-252: "The moisture content of cereal products must be less than 13% and it should be stored in polyethylene bags" Why should this be the case in Pakistan and perhaps other countries?

Round 3

Reviewer 1 Report

All comments have been adequately addressed.